# UAV-Based Wildland Fire Air Toxics Data Collection and Analysis

**DOI:** 10.3390/s23073561

**Published:** 2023-03-29

**Authors:** Prabhash Ragbir, Ajith Kaduwela, David Passovoy, Preet Amin, Shuchen Ye, Christopher Wallis, Christopher Alaimo, Thomas Young, Zhaodan Kong

**Affiliations:** 1Department of Mechanical and Aerospace Engineering, University of California, Davis, One Shields Avenue, Davis, CA 95616, USA; pragbir@ucdavis.edu (P.R.); pnamin@ucdavis.edu (P.A.); shcye@ucdavis.edu (S.Y.); 2Air Quality Research Center, University of California, Davis, One Shields Avenue, Davis, CA 95616, USA; apkaduwela@ucdavis.edu (A.K.); cdwallis@ucdavis.edu (C.W.); 3California Department of Forestry and Fire Protection, 715 P St., Sacramento, CA 95814, USA; david.passovoy@fire.ca.gov; 4Department of Civil and Environmental Engineering, University of California, Davis, One Shields Avenue, Davis, CA 95616, USA; cpalaimo@ucdavis.edu (C.A.); tyoung@ucdavis.edu (T.Y.)

**Keywords:** Unmanned Aerial Vehicles, wildfire, smoke plumes, air quality monitoring, low-cost sensors, volatile organic compounds

## Abstract

Smoke plumes emitted from wildland-urban interface (WUI) wildfires contain toxic chemical substances that are harmful to human health, mainly due to the burning of synthetic components. Accurate measurement of these air toxics is necessary for understanding their impacts on human health. However, air pollution is typically measured using ground-based sensors, manned airplanes, or satellites, which all provide low-resolution data. Unmanned Aerial Vehicles (UAVs) have the potential to provide high-resolution spatial and temporal data due to their ability to hover in specific locations and maneuver with precise trajectories in 3-D space. This study investigates the use of an octocopter UAV, equipped with a customized air quality sensor package and a volatile organic compound (VOC) air sampler, for the purposes of collecting and analyzing air toxics data from wildfire plumes. The UAV prototype developed has been successfully tested during several prescribed fires conducted by the California Department of Forestry and Fire Protection (CAL FIRE). Data from these experiments were analyzed with emphasis on the relationship between the air toxics measured and the different types of vegetation/fuel burnt. BTEX compounds were found to be more abundant for hardwood burning compared to grassland burning, as expected.

## 1. Introduction

Wildfires are one of the most destructive natural disasters and have the potential to cause significant damage to vegetation, property, and, most importantly, human life. In addition to the destruction caused by the fires themselves, the smoke plumes emitted produce air pollution, which deteriorates air quality [1,2,3]. In the United States, most wildfires occur in California, with 2020 being a record-breaking year of 8648 incidents and 4,304,379 acres burned. With these large numbers of incidents, the negative impacts of smoke plumes on human health are significant [4,5,6,7]. The fine particulate matter, carbon monoxide and carbon dioxide released into the atmosphere are known to cause respiratory problems, cardiovascular disease as well as other significant health problems in humans [8,9]. Furthermore, the fine particulate matter emitted from wildfires is predicted to increase towards the middle of this century [10]. Wildland Urban Interfaces (WUI) fires, in particular, are known to produce toxic air pollution due to the burning of synthetic components. Many of these toxic substances are included in the 188 hazardous air pollutants (HAPs) that the United States Environmental Protection Agency (EPA) has outlined. These air pollutants include volatile organic compounds (VOCs), some of which are toxic. For scientists to understand the impacts of these air toxics on human health, it is necessary for accurate and real-time data on the air toxics to be made available.

Currently, air quality is measured using ground-based sensors, manned airplanes, and satellites. However, these methods all provide low-resolution data. Ground-based sensors are sparsely located and do not provide a good representation of the air quality within smaller regions. Manned airplanes are costly to operate and require continuous forward flight, which makes them unsuitable for targeting very specific locations for sampling. Satellites can provide large spatial coverage; however, with a spatial resolution of kilometers and a temporal resolution of days, they cannot capture precise data in real-time. Unmanned Aerial Vehicles (UAVs) have the potential to provide high spatial and temporal resolution data due to their ability to hover in a fixed location and maneuver with precise trajectories in 3-D space. They are flexible in their usage and can be adapted to suit a wide range of applications by equipping them with the necessary sensors and actuators. Furthermore, multiple UAVs can form a UAV swarm and work together to complete complex tasks or tasks which require the coverage of a large area, such as in wildfire air pollution monitoring. Additionally, most UAVs utilize GPS data for navigation and flight control, and this data can be synchronized with air pollution data to visualize and better understand their spatial distribution. Octocopter UAVs, in particular, can transport relatively large and heavy payloads and are suitable for transporting multiple onboard systems, allowing for a wide range of sensors and other data collection devices to be flown. UAVs have been applied towards air quality monitoring in many different studies as outlined by [11,12]; however, their application towards wildfire air pollution monitoring is limited.

Light-weight and low-cost sensor packages for air quality measurements have been developed previously [13,14]. These include sensors for measuring particulate matter, carbon monoxide, and carbon dioxide as well as sensors for measuring atmospheric data such as temperature, humidity, and pressure. The sensors can be integrated into a single printed circuit board (PCB) and a microcontroller used to read and write the data to a storage device on the UAV or transmit the data to the ground using a wireless communication protocol. Many studies have been conducted using these types of sensors for air pollution monitoring in various environments. For example, UAVs were used to study the vertical profiles of various air pollutants such as fine particulate matter and black carbon by [15,16,17,18,19,20]. Additionally, Li et al. [21] studied both the vertical and horizontal spatial distributions of PM2.5 and Ozone in an urban environment using a UAV-based air pollution monitoring system. Gu et al. [22] also utilized air pollution sensors onboard a UAV to monitor PM2.5 concentration near vehicle and restaurant chimney emissions. An Internet-of-Things-based system that displays the time series data of CO, CO${}_{2}$, and air quality, obtained from a UAV during flight, was developed by [23]. Furthermore, Stojcsics et al. [24] mapped the CO pollution of a small city in 2D by surveying the area with a sensor array and then using GPS coordinates and interpolation to create the map. A similar approach was utilized by [25] where fine-grained 2D heat maps of the CO and NO${}_{2}$ concentrations at a university area, residential area, and an underground carpark were generated at various heights. Other technical capabilities to complement air quality monitoring, such as target detection and pollution abatement, were explored by [26,27], respectively. Recently, a study by [28] investigated the spatial and temporal distributions of air pollutants during and after a hurricane. Despite the wide range of studies conducted in the area of air pollution monitoring, the application of mapping the spatial distribution of wildfire smoke has been limited.

Although there exists a wide range of low-cost sensors, there are very few available for measuring organic compounds, and they are not very reliable. However, a well-established method for measuring VOC concentration is by using sorbent tubes for collecting air samples followed by an offline analysis of the samples using thermal desorption-gas chromatography-mass spectrometry (TD-GC-MS) [29]. These sorbent tubes are small and lightweight, and several of them can be mounted on a UAV to allow for multiple air samples to be collected during a single flight. The tubes are typically made of stainless steel or glass and are packed with solid adsorbent materials which trap traces of the chemical compounds from the air being sampled. A study by [30] measured VOC concentration in the Amazon rainforest using a UAV equipped with an air sampler that uses sorbent tubes. They also conducted CFD simulations to determine the best location to mount the air sampler to avoid the downwash of the propeller thrust from affecting the airflow into the air sampler. The center of the UAV’s undercarriage was found to be a suitable location. Their method was proven to be effective in identifying chemical compounds in the rainforest with a detection limit as low as 3 ppt. Additionally, Chen et al. [31] used a similar approach to monitor VOC concentrations near a landfill, refinery, and power plant. This method has been applied in our study to measure air toxics from wildfires, which has not been seen in the literature.

A study by [32] developed microfabricated gas preconcentrator chips for VOC sampling of air pollution and mounted them onboard a UAV. Their system was tested during an experimental fire they designed using a commercial charcoal grill to burn biomass and industrial substances. The smoke plumes emitted by the fire were sampled, and the chips were analyzed offline using GC-MS. It was stated that the chips could detect analytes down to 22 ppb with only 2 min of sampling time. They also sampled air pollution in the east Davis area, originating from the 2018 Camp Fire in Northern California. However, these samples were taken from a building and not during a UAV flight through the smoke plumes of the fire. Similarly, Aurell et al. [33] utilized a sensor and sampling system to measure and characterize the smoke plumes emitted from the burning of military ordinance and presented 3D spatial distributions of the CO${}_{2}$ concentration near the area of the plumes. Additionally, a needle trap sampler mounted onboard a UAV was utilized by [34] to track emission sources of industrial pollution, and Summa canisters were utilized by [35] to collect air samples near structural fires from characterizing the VOC concentrations. Although VOC monitoring using these various methods is well established and very effective, there has not been a focus on applying them towards characterizing wildfire smoke air toxics.

This study investigates the use of an octocopter UAV, equipped with a customized air quality sensor package and a VOC air sampler, for the purposes of collecting and analyzing air toxics data from wildfire plumes. A UAV prototype has been developed and successfully flight tested during several prescribed fires conducted by the California Department of Forestry and Fire Protection (CAL FIRE) in June and October 2022. The data from these experiments are presented and analyzed with emphasis on the relationship between the air toxics measured and the different types of vegetation/fuel burnt during the prescribed fires. While there have been many studies that demonstrate the method of UAV-based air quality monitoring, the main focus of this study is to differentiate between various fuels burnt in prescribed burn experiments by utilizing air toxics data collected from UAV-based air quality measurements. The overall goal of this research is to understand better the concentrations and distributions of the air toxics emitted by smoke plumes from various fuel sources, which is essential for understanding their effects on human health.

## 2. Materials and Methods

### 2.1. UAV Platform

The UAV platform is a custom built octocopter. It weighs 2.7 kg and has a payload capacity of 11 kg. The arms are foldable with lengths of 416 mm, and the height from the ground to the base plate is 391 mm. The flight controller used is the Cube Orange autopilot which is equipped with 3 redundant inertial measurement units (IMUs), i.e., accelerometers and gyroscopes, as well as 2 barometers and 1 magnetometer. The GPS used is the CubePilot Here3 which has a positioning accuracy of 2.5 m, and the telemetry radio used is the RFD 900+, which has a data transmission rate of 900 MHz and a range of >40 km. The UAV is powered by a 15 C, 22.2 V, 6 S, 22,000 mAh LiPo battery and has a working temperature of −10 °C to 40 °C. The total weight of the UAV, the air quality sensor package and the VOC air sampler are approximately 3.7 kg. The maximum flight time with this weight is approximately 20 min. A customized mounting case for the air quality sensor package and VOC air sampler has been designed, and 3-D printed using Polylactic acid (PLA) filament. The VOC air sampler is mounted above the base plate of the mounting case, and the air quality sensor package is mounted below, as shown in Figure 1a. All of the UAV’s electronics are mounted on top of the platform, as shown in Figure 1b.

The location for mounting the air quality sensor package and the VOC air sampler on the UAV was chosen to minimize the effects of the propeller downwash on the accuracy of the measurements. While we did not conduct our own experimental or CFD studies to validate this, the decision was based on previous studies by [30,36] who investigated these effects using experimental and CFD simulations, respectively. In both studies, it was shown that the center of the UAV’s undercarriage is a suitable location for mounting the instruments. The study by [36] also showed that a location off-center and away from the propellers is even more suitable. However, due to the long arm length of the UAV (1 ft 11 inches), mounting any instruments at this location would significantly affect the stability and control of the UAV. Therefore, it was justified to mount the measurement systems below the UAV.

### 2.2. Microcontroller

Both the air quality sensor package and the VOC air sampler use their own independent microcontrollers. The microcontroller used in both cases is the Spresense by SONY (Figure 2). The Spresense has a main board and an extension board. The main board has a CPU that includes 6 cores and has ultra-low power consumption. It also includes 1.5 MB of SRAM and 8 MB of flash memory. Devices such as the air quality sensors can be connected to the Spresense via GPIO, SPI, I2C, UART, or I2S digital input/output ports. The extension board has a micro-SD card slot which is useful for storing data collected from the sensors.

### 2.3. Air Quality Sensor Package

The air quality sensor package contains 6 low-cost air quality sensors and the Spresense microcontroller, which are all mounted on a customized PCB board. It also includes a real-time clock (RTC) for ensuring that the timestamp for each data point collected is known. This is especially important for syncing the sensor package data with the GPS data of the UAV. The Spresense runs a circuit python code for reading all of the sensor data and writing the data to a text file on the SD card mounted on the Spresense extension board. This code is provided in the Data Availability Statement section. Table 1 provides a list of the sensors used in the sensor package and their relevant specifications. The front and back views of the sensor package can be seen in Figure 3a and Figure 3b, respectively.

### 2.4. VOC Air Sampler

The VOC air sampler utilizes inert-coated stainless steel “Universal” thermal desorption sampling tubes (Markes International, Inc., Sacramento, CA, USA) which is packed with a proprietary mixture of sorbents. The sorbent tubes capture and store air samples for later offline analysis after data collection has been completed. Four sorbent tubes are used, three for sampling and one as a blank for measuring any background concentrations due to passive diffusion into the system. The air sampler uses a mini vacuum pump to provide the necessary air flow rate into the sorbent tubes and an inlet solenoid valve ensures that air only enters the system during sampling. A manual needle valve and airflow sensor are used to regulate the airflow rate, and this has been set to 150 sccm. The sampling of each of the 3 tubes is controlled using a solenoid valve manifold and a relay module. The logic for the sequence of sampling is written in C++ code that runs on the Spresense microcontroller and was programmed using the Arduino IDE. An external battery pack provides (via a power distribution board) both 12 V for all of the solenoid valves and 5 V for the microcontroller and the vacuum pump. Figure 4 shows the air sampler prior to being mounted on the UAV.

After the data collection experiments, the sorbent tubes were analyzed via thermal desorption-gas chromatography-mass spectrometry (TD-GC-MS). Desorption was performed using a Markes Unity 2 thermal desorption unit coupled to a Markes Ultra xr tube autosampler. Desorption was achieved by heating the tube to 250 °C for 5 min under a flow of helium and recollecting the sample onto a TO-15-specific focusing trap held at 20 °C. The focusing trap was then rapidly heated to 280 °C under a flow of helium which carries the analytes through a heated transfer tube and onto the head of the analytical GC column. Analysis was carried out on an Agilent 6890N gas chromatograph (Agilent Technologies, Inc., Santa Clara, CA, USA) coupled with an Agilent 5973 N single quadrupole mass spectrometer. Separation was achieved using an Agilent DB-VRX column (60 m × 0.25 mm, 1.4 μm film thickness) using helium as the carrier gas. The GC oven is held at 35 °C for 5 min before ramping up to 250 °C at 7.5 °C/min, with a final hold time of 10 min. Concentrations of the target compounds were calculated by comparison to a multi-point calibration curve prepared using a gaseous TO-15 analytical standard mix (Air Liquide USA LLC, Houston, TX, USA), with the gas drawn through a series of thermal desorption tubes for varying lengths of time to create the calibration curve. Figure 5 shows the equipment used for TD-GC-MS analysis of the sorbent tube samples.

### 2.5. Prescribed Fire Experiments

The UAV prototype has been tested during 4 prescribed fires conducted by CAL FIRE in the Northern California region. The first 2 experiments in June 2022 occurred at San Andreas and Pilot Hill. At this stage of the project, only the air quality sensor package was complete and mounted on the UAV. The San Andreas experiment focused on evaluating the performance of the UAV in the harsh conditions of the smoke plumes. The Pilot Hill experiment focused on data collection using the air quality sensor package, but air samples were taken on the ground using a prototype of the VOC air sampler since it was not mounted on the UAV at this stage of the project. The other 2 experiments were conducted in October 2022 at Angels Camp and the Jack London State Park. At this stage of the project, the VOC air sampler was completed and also mounted on the UAV, and data were collected using both systems. For each prescribed burn, the flight pattern for VOC sampling was a vertical flight hovering at three locations. The prescribed fire experiments allowed us to sample very different types of smoke plumes, and different flight patterns were explored to maximize the spatial coverage of the different plumes for the sensor package data collection. An example of the smoke plumes sampled is shown in Figure 6a,b. Further details on each of these experiments are outlined below.

#### 2.5.1. Pilot Hill

At the Pilot Hill experiment, an area of grassland was burned. In this experiment, the smoke plumes were small enough that the UAV could circle the entire plume multiple times during a single flight. This flight pattern allowed for a large sample area of the plume (Figure 7). The GPS data were then synchronized with the sensor package data to produce 2D and 3D spatial plots. This allows us to gain insights into the spatial distribution of the chemical compounds throughout the plume. Moreover, since the plume moved across the field at low altitudes, the VOC samples taken from the ground provide a good representation of the chemical content of the smoke plumes, and this data has also been included in the results section. The three samples were taken at the same location but at different times. The samples were taken at ground level just downwind of the plume and consecutively in 7 min intervals. With each sample, the smoke plume became less dense as it dispersed. The plumes were mixed with less dense white smoke, which is known to contain more water vapor, and darker smoke which is known to contain more air toxics.

#### 2.5.2. Angels Camp

For the Angels Camp experiment, 1500 acres of shrub and grass were burned. This experiment was particularly difficult to conduct since the prescribed fire took place at the base of a hill, and the UAV could only be launched from much further away at the top of the hill. This meant that much of the UAV’s energy was spent on getting to the plume’s location, and sample time was reduced. Additionally, ignition was propagated by a helicopter, and for safety, the UAV could only fly once the helicopter was grounded. By this time, the smoke plumes had become significantly less dense due to weather and vegetation conditions. Since the area of the smoke was too large to cover, the middle of the plume was sampled in a circular flight pattern followed by a vertical flight pattern from the base to the top of the smoke (Figure 8a). The VOC sampling was done through a vertical flight that stopped at three locations, the lower, middle, and upper parts of the smoke area, at altitudes of 20 m, 60 m, and 100 m (above ground level), respectively. The lower part of the smoke was mainly less dense white smoke and became significantly thinner with increasing altitude.

#### 2.5.3. Jack London

For the Jack London experiment, 15 acres of mixed hardwood were burned. The smoke plumes generated were large, far exceeding the maximum height that the UAV can be flown by regulation (400 ft). As a result, only a small portion of the plumes was sampled. To maximize the spatial coverage of the plume sampled, a flight pattern that continuously moved back and forth across the width of the plume at different heights was used (Figure 8b). Since the VOC sampling could not be done at the lower, middle, and upper parts of the plumes as in the case of Angels Camp, the 3 sampling locations were an area of less dense white plumes, an area of denser darker plumes, and an area just downwind of the plumes. These locations were at altitudes of 80 m, 120 m, and 40 m, respectively. Figure 6a shows the smoke plume generated during the prescribed fire at Jack London State Park. The UAV’s flight towards the smoke plume for sampling can be seen in Figure 6b. The data collected from all 3 prescribed fire experiments are presented in the next section.

## 3. Results

### 3.1. Air Quality Sensor Package Results

This section shows the data collected from the air quality sensor package. It should be noted that although low-cost sensors are known to not be very accurate, they do provide precise data. As a result, the data can be used in a relative manner for basic comparisons and to make preliminary judgments about the chemical contents of smoke plumes, as in this study.

Figure 9 shows 2D heat maps that were generated by synchronizing the UAV’s 2D GPS data with the sensor package’s chemical data. These plots were generated only for the Pilot Hill experiment since the entire perimeter of the plume was covered and a detailed image can be generated to illustrate the 2D spatial distributions of chemical concentrations across the smoke plume. Since only a small part of the much larger smoke plumes at Angels Camp and Jack London could be sampled, the heat maps were not generated for those experiments.

Figure 10, Figure 11 and Figure 12 show 3-D scatter plots that were generated by synchronizing the UAV’s 3D GPS data (Latitude, Longitude, and Altitude) with the sensor package’s CO${}_{2}$ and PM2.5 data from each of the prescribed fire experiments. For each of these plots, the spectrum ranges from 0 (blue) to the highest value detected for each individual experiment (red). The CO${}_{2}$ data are given in ppm and the PM2.5 data represents the number of particles detected.

### 3.2. VOC Air Sampler Results

Figure 13 shows the VOC data that was generated via the TD-GC-MS process for the air samples collected at each experiment site. Each bar in the graph represents 1 of the 3 samples that were taken at each experiment site. The compounds detected are on the *x*-axis and the quantities are given in nanomoles on the *y*-axis.

Figure 13a shows the samples taken at Pilot Hill. Tubes 1, 2, and 3 were sampled at the same location but at different times starting from tube 1 and increasing in 7 min intervals. With each sample, the smoke plume became less dense and the corresponding lower concentrations of the VOCs can be observed.

Figure 13b shows the samples taken at Angels Camp. Tubes 1, 2, and 3 were sampled at three different locations, the lower, middle, and upper parts of the plume, respectively.

Figure 13c shows the samples taken at Jack London. Tubes 1, 2, and 3 were sampled at the less dense white part of the plume, the denser darker part of the plume, and a location downwind of the plume, respectively.

## 4. Discussion

Using a novel combination of a VOC air sampler and a customized air quality sensor package mounted on a UAV, we could spatially locate the different chemical compounds and their concentrations in multiple prescribed burns. CO${}_{2}$ is the most common emission factor produced by wildland fuels. At the Pilot Hill burn, CO${}_{2}$ rapidly dispersed, as seen in Figure 9a, where concentration is over 800 ppm on the right side of the image (flaming front of the fire) to less than 300 ppm on the left side of the image. At Angles Camp Figure 11a, also a grass and forb fuel type, the CO${}_{2}$ concentration was 750 ppm nearest the flaming front but decreased the higher in the column was sampled. Prescribed fires are optimal when there is a breeze to help with smoke dispersion and flaming consumption. It is likely that at Pilot Hill, the wind was faster and, thus, able to disperse the CO${}_{2}$ faster than the heat from the fire as compared to at Angles Camp. The CO${}_{2}$ concentration at Jack London was mainly around 600 ppm with some higher areas having over 950 ppm Figure 12a, this is to be expected given the heavier fuel load in mixed hardwood and based on results from SERA [37].

There were less than 100 PM2.5 particles detected throughout the plume at Pilot Hill, as shown in Figure 9b. However, at the location where the UAV took off and landed, the PM2.5 count increased significantly to over 5000. This was likely due to the UAV taking off from a dirt track and the thrust from the propellers causing a drastic rise in the number of particles and not from the smoke itself. This can be verified using the 3D spatial plot in Figure 10b that shows the high PM2.5 count near the ground where the UAV started its flight. There was generally a higher PM2.5 count at Angels Camp (400 to 600), with the highest being around 1200 at a very small area in the lower part of the plume (Figure 11b). At Jack London, the PM2.5 count at the area sampled was at most 450 (Figure 12b).

As shown in Figure 13a, at Pilot Hill, the VOCs that were present in the highest concentrations were 3-Chloropropene, Benzene, and Toluene at around 7, 9, and 8 nanomoles, respectively. At Angels Camp (Figure 13b), the highest concentrations were 1,2-Dichloroethane and 1,4-Dichlorobenzene at 0.03 and 0.017 nanomoles, respectively. At Jack London (Figure 13c), the highest were Benzene, Toluene, m,p-Xylene, and o-Xylene at 0.6, 0.35, 0.5, and 0.3 nanomoles respectively. At Pilot Hill, the samples were taken at the same location but at different times as the smoke plume moved past the air sampler. The decrease in concentrations with each sample is due to the smoke plume becoming thinner with time as it dispersed. At Angels Camp, the sampling locations were the lower, middle, and upper parts of the smoke area. Some of the compounds show a slight decrease in concentrations with an increase in height which is expected since the plumes become thinner with height. At Jack London, the 1st sample at the thicker part of the plume showed higher concentrations than the 2nd sample at the thinner part of the plume. The 3rd sample was taken away from the plume at a downwind location where the concentrations of VOCs were expected to be significantly less, as observed in the results.

One scientific question we try to answer is if we can distinguish between different fuel types in these prescribed burns based on individual measurements. That is, can we identify marker species for different fuel types? The immediate answer is no because we have only selected compounds that were measured. To answer this question fully, one has to measure a comprehensive suite of chemical compounds, which we intend to do in the future. The top two panels of Figure 13 are for VOCs (or HAPs) generated during two grassland-prescribed fires, and the bottom panel is for a mixed hardwood fire. In the top two panels, the abundance of compounds is somewhat similar, but in the bottom panel, some compounds are more abundant than others. These are mainly the BTEX (Benzene, Toluene, Ethylbenzene, and Xylenes) compounds with a few long-chain aliphatic compounds. BTEX is prevalent during wildfires and is also present in automobile emissions. This may suggest a relative way to identify hardwood fuel burning compared to grass burning. We did not see any distinguishing features in inorganic gas spectra.

## 5. Conclusions

Characterizing the air pollution emitted from a wildfire smoke plume requires high spatial coverage as well as the ability to sample at specific locations, which UAV-Based monitoring can provide. The air quality sensor package and VOC air sampler mounted on the UAV platform allowed for many different air toxins to be measured. The system was tested in several controlled fire experiments, each providing a different size and type of plume to sample. Different flight patterns suitable for the particular plume being sampled were explored, and 2D and 3D plots were generated for a better understanding of the spatial distributions of the air toxics within the plumes. The study focused on analyzing CO${}_{2}$, PM2.5, and VOC data, and the concentrations of each of them were correlated to the nature of the plume being sampled and the type of vegetation that was burned. The next step in this work will be to expand the system into a UAV swarm so that the spatial distribution of air toxics can be better captured and understood. Furthermore, future work will also include using deep learning to develop a pollutant propagation model using the air toxics data collected from the UAV swarm.

## Figures and Tables

**Figure 1 sensors-23-03561-f001:**
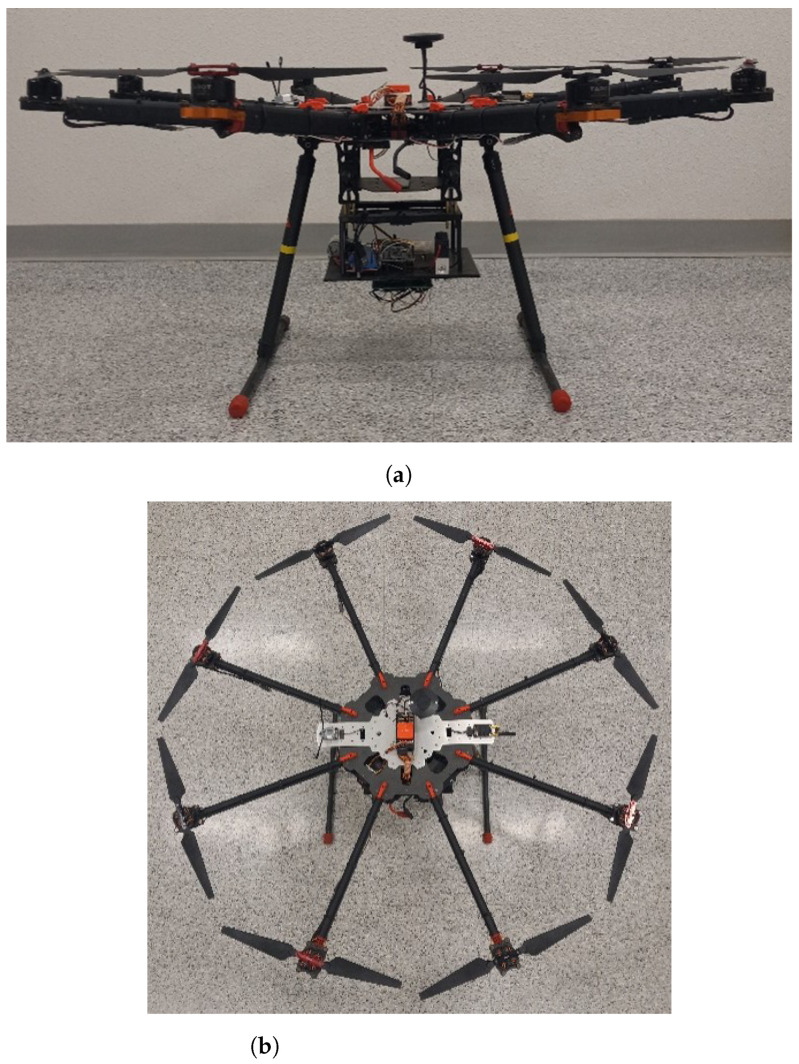
(**a**) Front view and (**b**) top view of octocopter UAV platform equipped with air quality sensor package and VOC air sampler.

**Figure 2 sensors-23-03561-f002:**
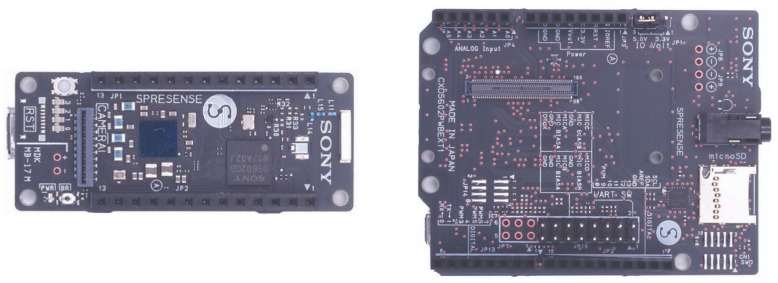
The main board (**left**) and extension board (**right**) of the Sony Spresense microcontroller used for the air quality sensor package and VOC air sampler.

**Figure 3 sensors-23-03561-f003:**
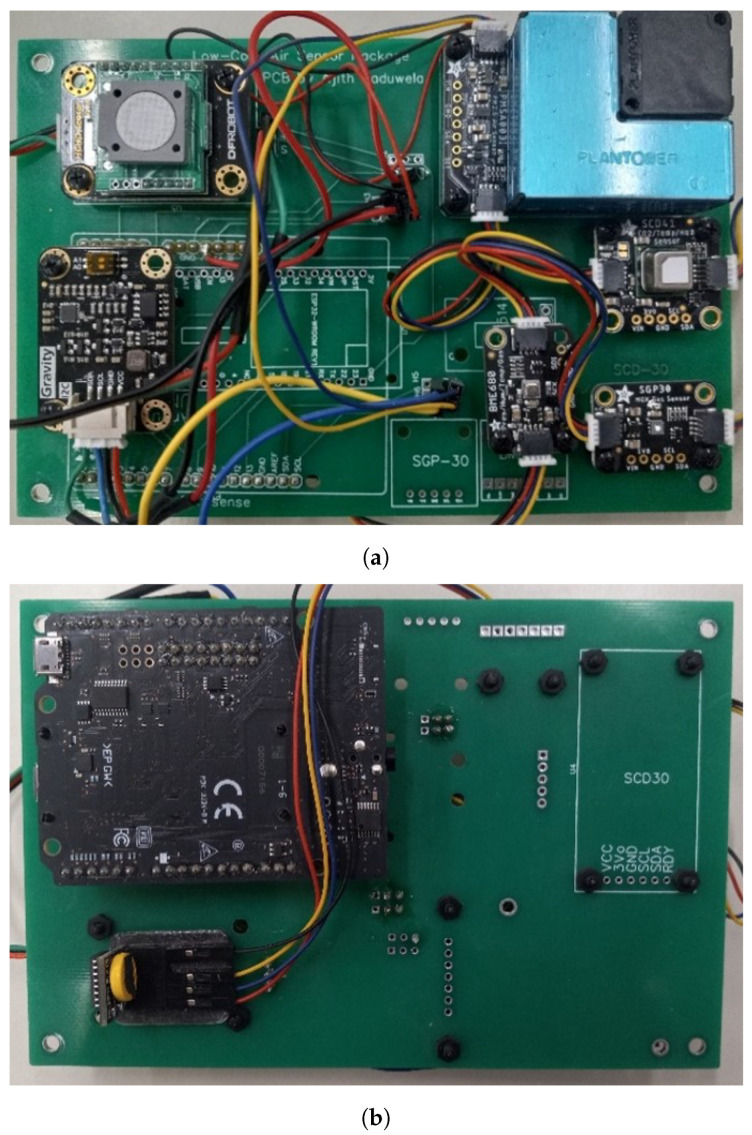
(**a**) Front view and (**b**) back view of the air quality sensor package.

**Figure 4 sensors-23-03561-f004:**
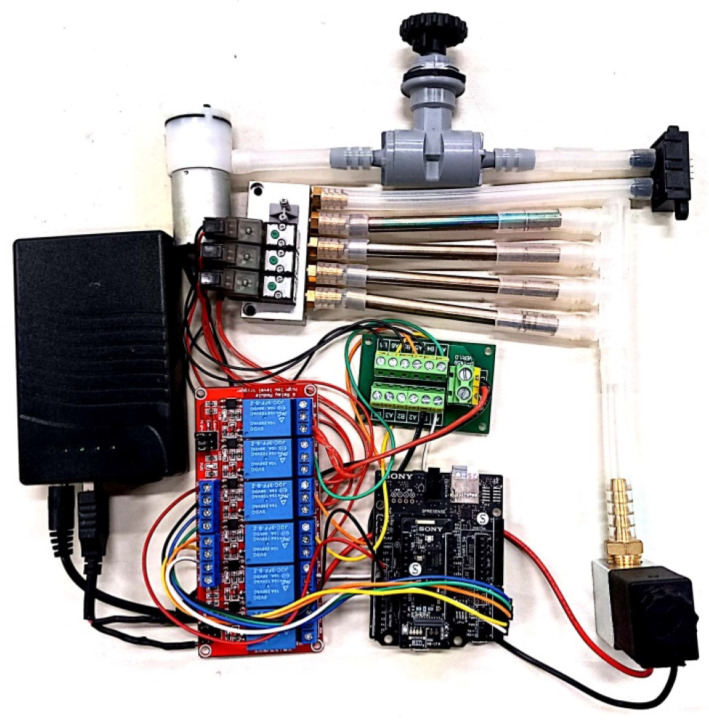
VOC sorbent tube air sampler.

**Figure 5 sensors-23-03561-f005:**
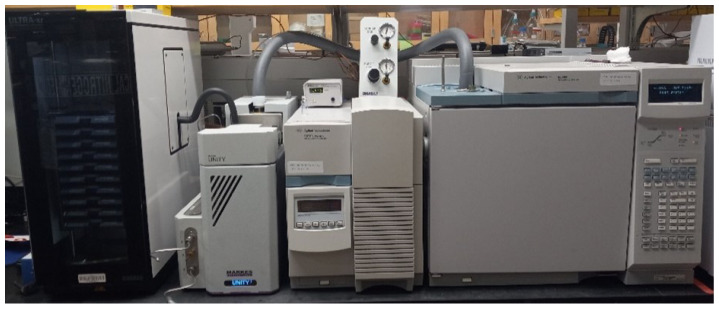
Equipment used for Thermal Desorption-Gas Chromatography-Mass Spectrometry analysis.

**Figure 6 sensors-23-03561-f006:**
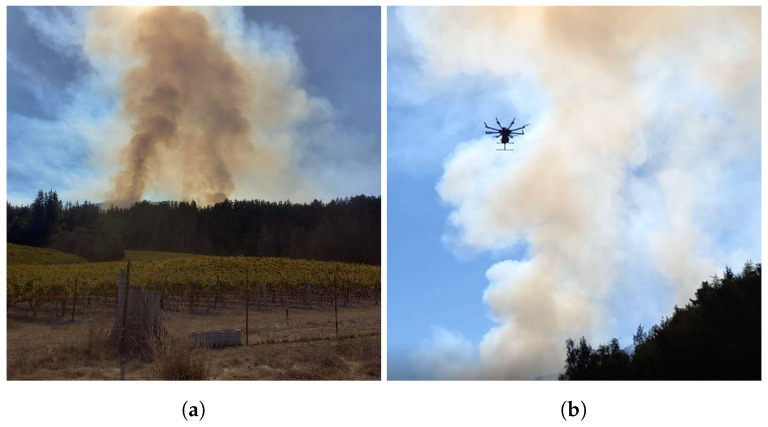
(**a**) Smoke plume emitted from a prescribed fire at Jack London State Park. (**b**) UAV-based data collection during a prescribed fire at Jack London State Park.

**Figure 7 sensors-23-03561-f007:**
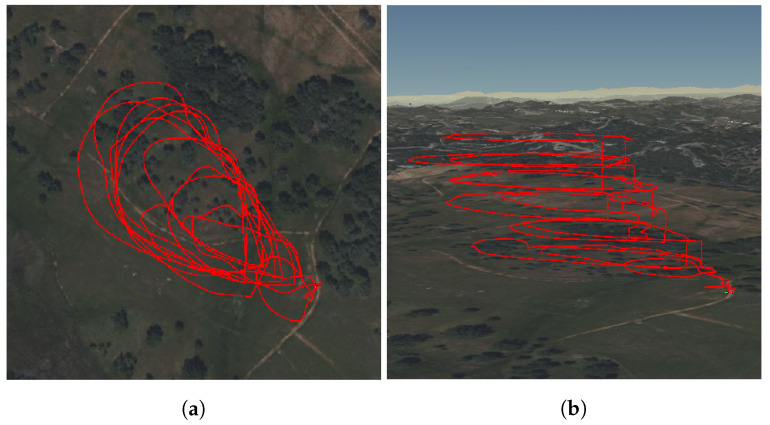
(**a**) 2D GPS data and (**b**) 3D GPS data for UAV data collection during a prescribed burn at Pilot Hill.

**Figure 8 sensors-23-03561-f008:**
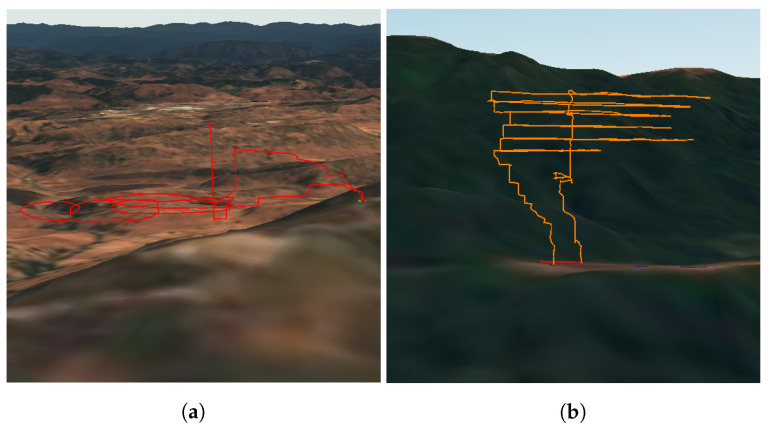
3D GPS data for (**a**) Angels Camp and (**b**) Jack London Prescribed Fire Experiments.

**Figure 9 sensors-23-03561-f009:**
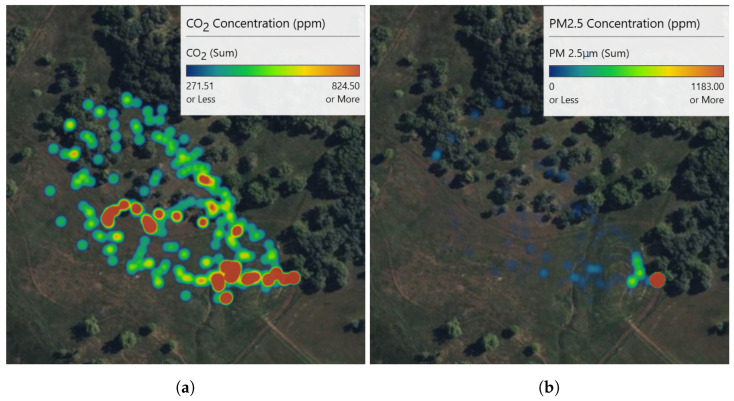
2D Heat Maps for (**a**) CO${}_{2}$ and (**b**) PM2.5 from the Pilot Hill Prescribed Fire Experiment.

**Figure 10 sensors-23-03561-f010:**
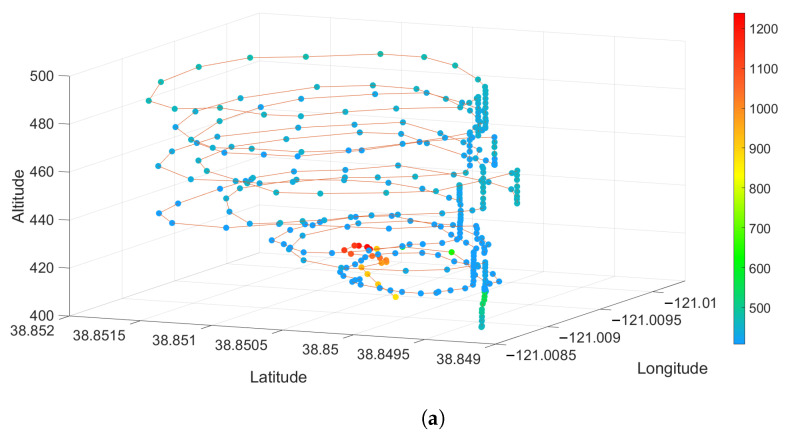
3D Scatter Plot for (**a**) CO${}_{2}$ and (**b**) PM2.5 from the Pilot Hill Prescribed Fire Experiment.

**Figure 11 sensors-23-03561-f011:**
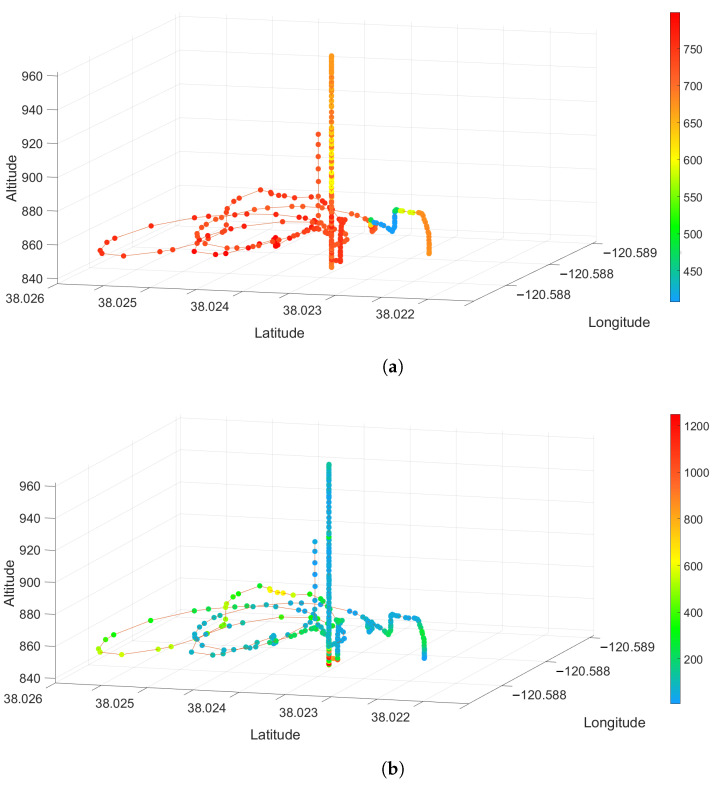
3D Scatter Plot for (**a**) CO${}_{2}$ and (**b**) PM2.5 from the Angels Camp Prescribed Fire Experiment.

**Figure 12 sensors-23-03561-f012:**
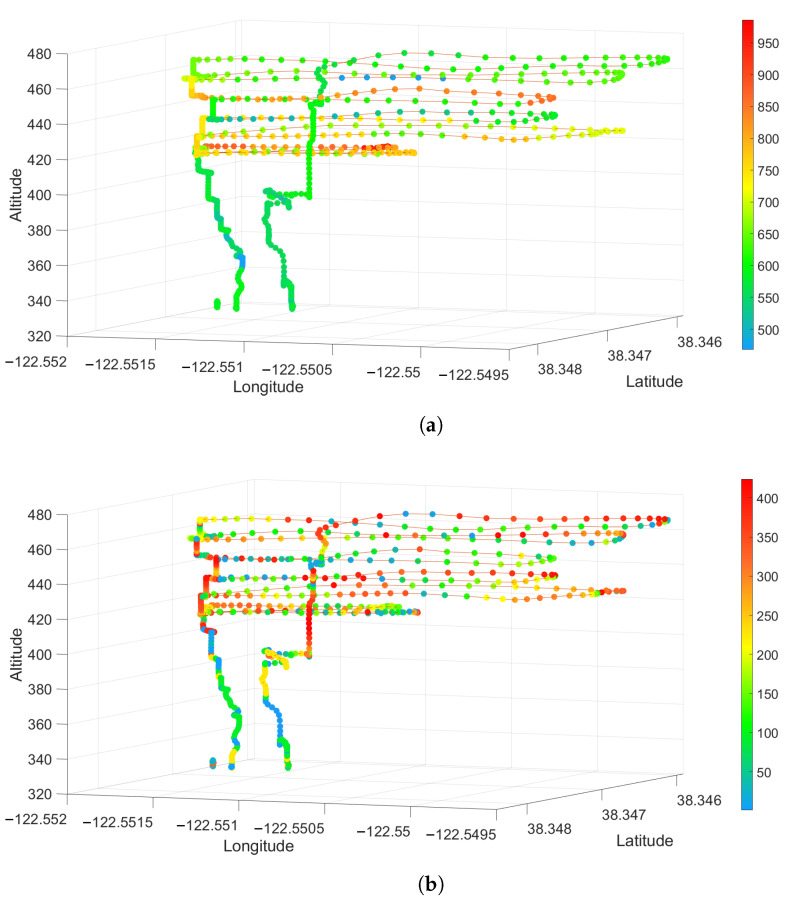
3D Scatter Plot for (**a**) CO${}_{2}$ and (**b**) PM2.5 from the Jack London Prescribed Fire Experiment.

**Figure 13 sensors-23-03561-f013:**
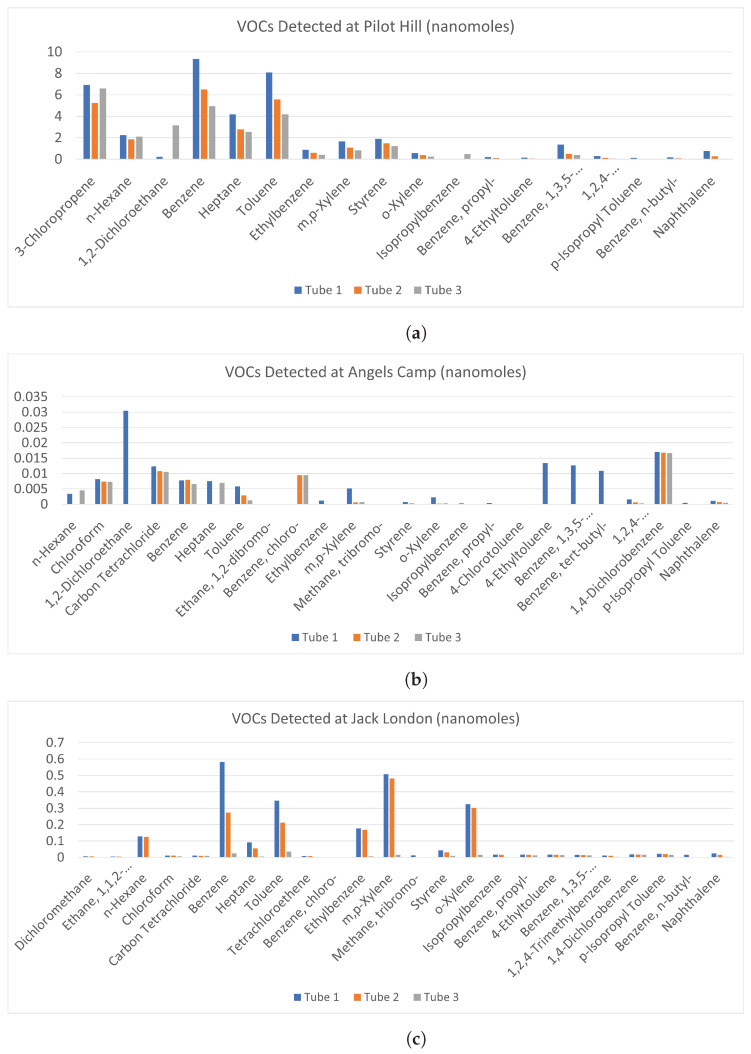
VOC Data for the (**a**) Pilot Hill, (**b**) Angels Camp, and (**c**) Jack London Prescribed Fire Experiments.

**Table 1 sensors-23-03561-t001:** Air quality sensors used for the sensor package and their specifications.

Sensor	Measurement	Operating Range
BME680	Temperature	−40 to 85 °C
	Barometric Pressure	300 to 1100 hPa
	Relative Humidity	0 to 100% r.H.
SGP30	eCO${}_{2}$	400–60,000 ppm
	Total VOC	0–60,000 ppb
SCD41	CO${}_{2}$	400–5000 ppm
PMSA003I	Particulate Matter	0–500 μg/m${}^{3}$
SEN0231	Formaldehyde (HCHO)	0–5 ppm
MiCS-4514	CO	1–1000 ppm
	NO${}_{2}$	0.05–10 ppm
	NH${}_{3}$	1–500 ppm

## Data Availability

Data supporting reported results can be found at https://github.com/CHPS-Lab/uav_wildfire (accessed on 10 February 2023).

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
