# Peer review of "UAV-Based Wildland Fire Air Toxics Data Collection and Analysis"

_sensors, 2023, doi:10.3390/s23073561_

Round 1

Reviewer 1 Report

The topic covered by the authors in this article is interesting, but the authors are probably in an early stage of their research. In fact, it is not clear which scientific contribution the authors want to highlight, since the authors present the results of the acquisitions made with low-cost sensors installed on board a drone. The reports of the acquisitions synchronized with the GPS positions of the drone in particular test areas are then presented. In its current form the article cannot be published. First of all because if the authors intended to present an instrument for measuring the concentrations of gases in the air produced by a fire, the sensors used do not represent an acceptable solution. In fact, it is known that the sensors used do not provide a quantitatively reliable measurement of the concentration of the gases measured, and their calibration, even repeated over time, would be at least advisable. If, on the other hand, the purpose of the research consisted in the study of the burnt material from the analysis of the fumes, then a significant part of the research that must be carried out is missing, probably also making use of artificial intelligence algorithms.

Reviewer 2 Report

The research topic of this study presents that (1) a customized air quality sensor package and VOC air sampler were mounted on an octocopter UAV for autonomous flight, (2) harmful substance data from smoke plumes were collected, and (3) an analysis was conducted according to the materials (vegetation/fuel). However, the following issues were identified in the study:

1. There is no explanation regarding the effect of the propellers on the surrounding air caused by their operation, the impact on the sensor, and the accuracy of the flight.

2. Although graphs were created to visualize and data-ize the smoke plumes (e.g. figures 7, 8, 10, 11, 12), an explanation regarding the errors of these graphs is necessary. While the data was used to create graphs and statistics, the study lacks detailed data tables, analysis techniques, and justifications.

3. The drone was flown in various forms in different experiments to collect samples from the lower, middle and upper part of the smoke area, and thus a precise description of the locations is required. Additionally, with only three samples collected for each experiment, the persuasiveness of the data analysis is limited. While the author mentioned the lack of persuasiveness in the paper, they still conducted an analysis, and an explanation for this is necessary.

4. Due to the lack of data, it is considered meaningless to evaluate the harmfulness of smoke plumes. Therefore, to generate credibility for the analysis, a sufficient amount of reliable data is necessary.

5. The study aimed to differentiate various combustion substances. However, the reliability of the data indicating that BTEX compounds are more abundant during the combustion of hardwood compared to grassland is questionable due to the limited amount of data. To generate credibility for the analysis, a sufficient amount of reliable data is necessary.

Reviewer 3 Report

Dear Authors, 

first of all thank you very much for the opportunity to read your paper. I have few small suggestion. 
Figure 7,8 maybe is possible to add metric scale,

subsection "3.2. VOC Air Sampler Results" includes only plots without text, usually subsection should include both. I suggest to remove subsection or add some text.

FigurÄ™ 10-12 - bigger units. 

Best regards, 

Round 2

Reviewer 2 Report

The author carefully reviewed the opinions I presented and corrected the suggestions accurately. Therefore, the passage of this thesis is approved.